# A Feature Analysis Based Identifying Scheme Using GBDT for DDoS with Multiple Attack Vectors

**Jian Zhang, Qidi Liang, Rui Jiang and Xi Li \***

School of Computer Science and Engineering, Central South University, Changsha 410083, China;
csu_jianzhang@263.net (J.Z.); liangqidi@csu.edu.cn (Q.L.); ruijiang@csu.edu.cn (R.J.)

**\*** Correspondence: lixi@csu.edu.cn

**Abstract:** In recent years, distributed denial of service (DDoS) attacks have increasingly shown the trend of multiattack vector composites, which has significantly improved the concealment and success rate of DDoS attacks. Therefore, improving the ubiquitous detection capability of DDoS attacks and accurately and quickly identifying DDoS attack traffic play an important role in later attack mitigation. This paper proposes a method to efficiently detect and identify multivector DDoS attacks. The detection algorithm is applicable to known and unknown DDoS attacks.

**Keywords:** traffic characteristics; DDoS detection; feature selection; GBDT algorithm; attack feature tree

---

## 1. Introduction

Today, an ever-increasing number of businesses are using data centers or large server clusters to run a variety of applications. Most of these applications use TCP(Transmission Control Protocol) or UDP(User Datagram Protocol) protocols, such as web services that make up most of the network's traffic. However, applications based on TCP or UDP have suffered a variety of malicious attacks, especially distributed denial of service (DDoS) attacks. Denial of service (DoS) or DDoS attacks pose a devastating threat to network services [1]. In the field of network security, detecting, identifying, and mitigating denial of service and distributed denial of service attacks is a challenging task [2]. For the current, widespread Mixed DDoS attacks, detection is mainly used to determine whether the current traffic has DDoS attack behavior, and identification is used to provide decision information regarding various specific attack types. At present, researchers have proposed many abnormal traffic detection models, including feature matching, statistical rules, and machine learning. These models [3] are widely used for abnormal flow monitoring. In recent years, using machine learning to detect abnormal traffic has become a hot spot for DDoS traffic detection. Marwane Zekri et al. [4] proposed a DDoS detection system based on the C4.5 algorithm to mitigate the DDoS attack threat. Wathiq Laftah AY et al. [5] proposed a multilevel hybrid intrusion detection model that uses support vector machines and extreme learning machines to improve the efficiency of detecting known and unknown attacks. Kuang et al. [6] proposed an intrusion detection system based on an SVM(Support Vector Machines) model combined with kernel principal component analysis (KPCA) and a genetic algorithm (GA). However, the current research on DDoS attack detection has the following problems: (1) more consideration is given to the detection of DDoS attack behavior, but less recognition exists for composite type attacks; (2) the type of attack identified by the classifier using limited statistical features is a limitation; (3) the connection between the attack feature and the attack type cannot be given; and (4) the DDoS attack detection accuracy is not high, and the detection response time is still slow.

In recent years, many machine learning methods have been developed to detect abnormal traffic, but most researchers use the KDD-Cup 99 [7] dataset and the traffic characteristics provided by KDD-Cup 99 to experiment, but KDD-Cup 99 provides a limited number of DDoS attack types.

---

Alternatively, these researchers only apply a small amount of traffic characteristics. Practical experience has shown that the data used and their characteristics determine the effectiveness of machine learning, the algorithm and algorithm optimization simply approximate the obtained result of machine learning. Based on TCP traffic characteristics in 102 and 49 UDP traffic characteristics, this paper constructs a feature subset that accurately represents different types of attacks through a feature selection algorithm and then optimizes the parameters of the GBDT algorithm to achieve an accurate and fast identification of the purpose of malicious attack traffic in TCP and UDP flows. To effectively shorten the training and detection time of the algorithm and improve the ubiquitous ability of the algorithm, we use the combination of random forest and Pearson correlation coefficient as the search strategy, and use the GBDT algorithm as the evaluation standard to implement the feature selection algorithm. Based on this approach, the GBDT algorithm is tuned to identify DDoS composite attack types. The experimental results show that the GBDT algorithm can quickly and accurately identify DDoS attack traffic after feature selection and tuning.

This article contributes the following:

(1) From the thousands of features used in the DDoS traffic detection literature in recent years, 102 features are extracted and implemented as the original feature set of DDoS attack recognition. These features can effectively represent known and unknown DDoS attacks. The experimental results show that using these features on the 2017 WIDE dataset can accurately detect DDoS attack traffic;

(2) A feature selection method based on the random forest feature score and Pearson correlation coefficient is proposed. The method is compared with the traditional dimensionality reduction algorithm and feature selection method. The experimental results show that the method utilizes a smaller feature subset to maintain or improve the original attack detection accuracy;

(3) To identify the various DDoS attacks in the composite DDoS attack traffic completely, accurately and quickly, an attack identification algorithm based on GBDT is proposed, and the GBDT parameter optimization method is given. In addition, the nearest neighbor, Bayesian, and support vector machine are compared with machine learning algorithms such as a deep network, and the experimental results show that the GBDT algorithm is superior to other algorithms in attack type recognition accuracy and running time.

(4) The method of constructing and optimizing the attack feature tree is proposed and continuously optimizes the attack feature to characterize the specific type of attack by analyzing the relationship between the attack type and the attack feature.

The organizational structure of this paper is as follows. The second part discusses the related work and summarizes researchers' results in existing work. The third part introduces the DDoS attack feature selection framework and mode. The fourth part introduces the DDOS attack detection algorithm based on GBDT and its classification and the performance comparison experiment between other machines and other machine learning algorithms. The fifth part introduces a feature selection algorithm based on random forest and Pearson correlation coefficient analysis and its optimization for the DDoS classifier. The sixth part introduces the parameter tuning of the GBDT algorithm, the feature selection algorithm and its application in the performance improvement of DDoS attack identification. The seventh part summarizes the entire paper.

## 2. Background and Related Work

Feature extraction is a very important step in the machine learning process. The quality of the features determines the pros and cons of machine learning. The third International Knowledge Discovery and Data Mining Competition (KDDCUP'99) provides 41 traffic characteristics for abnormality detection, such as duration, protocol_type, and service in the intrusion detection datasets [7]. Traffic statistics are very important for DDoS detection. Jiahui Jiao [8] et al. defined two attack modes, fixed source IP attack and random source IP attack, and proposed a DDoS real-time detection method

based on TCP protocol. This method extracts 15 basic statistical features and 16 ratio characteristics of TCP traffic. Malicious traffic is distinguished from normal traffic by two decision tree classifiers. Qin.X [9] proposed an entropy-based DDoS attack detection method. By constructing the entropy vector of different traffic characteristics, the clustering analysis algorithm is used to model the normal mode. By defining different packet size levels, different levels of packets are set to different characteristics. It chooses the network connection quintuple (source IP, destination IP, source port, destination port, protocol number), and the packet size (divided into five levels, such as 0–128, 128–256, 256–512, 512–1024, 1024–1500) and $\frac{SYN}{SYN+ACK}$ for a total of 11 features that are used for attack detection. Andrew W. Moore et al. [10] proposed 249 features in traffic classification, including link layer features, IP layer features, and TCP layer features. This method defines the maximum, minimum, 1/4, 1/2, and average values of the packet size (or arrival time) from the client to the server and from the server to the client. The features are closely related to traffic classification, and thus indicate that these features can be applied to TCP, UDP, and ICMP flow anomaly detection. Yaokai Feng et al. [11] detected DDoS attacks using 55 features, such as minimum packet size, average size, and size variance of packets in the session, and selected 10 features out of 55 features to characterize ChallengeCoHapsar (CC) attacks. The CC attack is a type of DDoS attack that uses a proxy server to send a large number of seemingly legitimate requests to the victim server. CC is named according to its tools, and the attacker uses a proxy mechanism to launch DDoS attacks using a number of widely available free proxy servers. Many free proxy servers support anonymous mode, which makes tracking very difficult.

Feature selection is an important means to improve the accuracy of machine learning detection and shorten the detection time. Excessive features will result in redundancy, and feature redundancy will increase the time required for machine learning training models; consequently, the detection accuracy will decrease, and the detection time will increase. Therefore, many researchers are committed to the study of feature selection. Yaokai Feng et al. [11] considered that important features are essential for early detection of DDoS attacks, and use support vector machines (SVM) and principal component analysis (PCA) as feature selection algorithms. Their experimental results show that there are 10 features that can characterize CC attacks. Liu et al. [12] proposed the use of mutual information methods to eliminate redundant features. Table 1 summarizes the feature selection methods that are currently used by researchers in DDoS testing.

Machine learning algorithms are the core of machine learning. Different algorithms can produce different detection accuracies and detection times when they act on the same datasets. Detection accuracy and detection time are two major indicators that directly affect the DDoS detection effect. In recent years, many researchers have often used machine learning algorithms to detect abnormal traffic, and the detection results they obtained are also inconsistent. Common machine learning algorithms include principal component analysis, K-nearest neighbor (KNN), naive Bayes classifier (NB), decision tree, support vector machine, K-means (K-means), and back propagation neural network (back propagation, BP).

The PCA-based DDoS attack detection was first proposed by A. Lakhina et al. [13]. The method separates the high dimensional space occupied by a set of network traffic measurements into disjoint subspaces corresponding to normal and abnormal network conditions. Experiments have shown that this separation can be effectively achieved by principal component analysis. There are many differences between this approach and the PCA-based multivariate statistical process control (MSPC) method in the industrial processing and chemometric literature. José Camacho et al. [14] recommended the use of multivariate statistical network monitoring (MSNM) to effectively avoid the shortcomings mentioned in the literature, and the limitations of using PCA in the network are reported.

**Table 1.** Feature selection method.

| Author | Feature Selection Method | Authentication Method | Datasets |
| --- | --- | --- | --- |
| Wei W. et al. [15] | Information gain and chi-square method | Bayesian network | KDD-Cup 99 |
| Fatemeh A. et al. [16] | Linear correlation coefficient and forward feature selection algorithm (FFSA) and proposed modified mutual information feature selection (MMIFS) | SVM, Bayes | KDD-Cup 99 |
| Yinhui Li et al. [17] | Stepwise feature removal method GFR method | SVM | KDD-Cup 99 |
| Jarrah [18] | Random forest-forward selection sort (RF-FSR) and Random forest-backward sorting (RF-BER) | Decision tree, forest | KDD-Cup 99 |
| O.Y. AlJarrah [19] | Consistency subset evaluation | Extreme learning machine | NSL-KDD |
| Ay [20] | CFS Subset Evaluator is used as an attribute evaluator and best first is used as a search method | Random Forest, J48, and Naive Bayes | NSL-KDD |
| R. Vijayanand [21] | Genetic algorithm multisupport vector machine | SVM classifier | CICIDS 2017 |
| Tarfa Hamed [22] | Recursive feature addition (RFA) and bigram techniques | SVM classifier | ISCX |
| Chaouki K. [23] | Genetic algorithm as feature search and logistic regression as a packaging method for learning algorithms | Decision tree | KDD-Cup 99, UNSW-NB15 |
| Yang Li [24] | Improved linear SVM | SVM | KDD-Cup 99 |

Based on KNN-based DDoS attack detection, Ming-YangSu [25] proposed a method for detecting large-scale DDoS attacks in real time by a weighted KNN classifier. In addition, a combination of a KNN and a genetic algorithm for feature selection and weighting is proposed. The experiment shows that the overall accuracy is as high as 97.42% for known attacks. For unknown attacks, a 78% accuracy rate was obtained. Known attacks and unknown attacks are relative to the training set of machine learning. If the training set contains a certain DDoS attack type, the DDoS attack type belongs to a known attack. If the DDoS attack type is not included in the training set, the DDoS attack type belongs to an unknown attack. If a model can detect not only known attacks but also unknown attacks, the model performs very well. When multiple DDoS attacks occur simultaneously, the ability of the model to detect unknown attacks is very important. The authors in [26] proposed a hybrid approach to intrusion detection systems that uses a boundary-cutting algorithm of the Manhattan and Jaccard coefficients with similar distances that was combined with the KNN algorithm to implement intrusion detection.

Based on NB-based DDoS attack detection, Yunpeng Wang et al. [27] applied the NB and ReliefF algorithms to propose a naive Bayesian classification method. It uses the ReliefF algorithm to assign a weight to each attribute in the KDD-Cup 99 datasets, which reflects the relationship between the attribute and the final class for better classification results. Thaseen, S, I et al. [28] proposed an intrusion detection model using linear discriminant analysis (LDA), chi-square feature selection, and improved naive Bayesian classification. They use Bayesian classifiers to identify normal and abnormal traffic in the NSL-KDD datasets. The experimental results show that the Bayesian classifier combined with other feature selection methods yields higher accuracy and a lower false positive rate.

Regarding decision tree-based DDoS attack detection, Lakshmi et al. [29] use decision trees to protect wireless nodes and target nodes within the network from DDoS attacks. Malik, A, J et al. [30] used standard particle swarm optimization algorithms to prune the decision tree with single and multiple target perspectives. The pruned decision tree classifier is then used to detect anomalous network connections. Experiments on the KDD-Cup 99 datasets showed an average detection accuracy of 93.26%.

SVM-based DDoS attack detection has been implemented. Adel. A. et al. [31] proposed an SVM-based framework for detecting denial of service attacks in virtualized clouds in a changing infrastructure that collects some systems. The metrics are used to train SVM classifiers to distinguish between normal and malicious activities of virtual machines (VMs), associating VM application metrics with actual resource loads that enables hypervisors to distinguish between a benign high load and DDoS attacks. Wang et al. [32] proposed an effective intrusion detection framework based on enhanced

feature support vector machine (SVM). The logarithmic edge density ratio is transformed to form the original features, such that new and better quality transform features are obtained and the SVM-based detection model can achieve greatly improved detection ability. Feng, W, Y et al. [33] combined the SVM method with the self-organizing ant colony network cluster (CSOACN) to take full advantage of both approaches, and achieved an accuracy of 95.3% on the KDD-Cup 99 datasets.

Approaches based on K-means DDoS detection have been employed. Gina C [34] fully integrated a SOM neural network and the K-means algorithm, and developed a two-stage classification system to correlate related alarms and further classify alarms. This method effectively reduces all redundant and noisy alarms. The author used the DARPA dataset to conduct experiments. The results show that the method can detect 96% of the false positives, and the false positive rate is much lower. Ujwala Ravale et al. [35] proposed a hybrid technique that combines the K-Means clustering algorithm and the RBF kernel function of the support vector machine as a classification module. The main goal of the proposed technique is to reduce the number of attributes associated with each data point. The accuracy of the proposed technique in the KDDCUP'99 datasets reached 96.3%.

BP-based DDoS attack detection was reported in the following work. Zouhair Chiba et al. [36] proposed an optimal method for constructing anomalous NIDS(Network Intrusion Detection System) based on back propagation neural network using a back propagation learning algorithm. Yunhe Cui et al. [37] used BP to detect traffic in real time on a software defined network (SDN). They first obtained all the flow entries for each switch. After receiving the flow statistics of the flow entry sent by the switch, the controller parses the flow statistics and processes the flow entry in the message one by one. The eigen values of the flow entry include the number of packets matched by each flow entry, the number of bytes matched by each flow entry, the lifetime of each flow entry, the packet rate of each flow entry, and the byte rate of each flow table item. The extracted feature values are then passed to the BP to determine if the TCP or UDP flow is benign or malicious. This method of detection takes a long time.

At present, composite type attacks in DDoS detection are little recognized, and the limited statistical features implemented in classifiers result in limited types of attacks. Moreover, the detection accuracy of DDoS attacks is not high, and the detection response time is still slow. Therefore, this paper also provides the connection between attack characteristics and attack types, helping decision makers quickly lock the scope of attack types.

## 3. System Model and Problem Statement

The overall framework of this paper is shown in Figure 1. The first step is to extract the data from the pcap file, that is, parse the data packets of each pcap file. Then, the data packet is synthesized into a TCP flow or a UDP flow from which features are extracted. Finally, each datum is tagged, and one datum is formed by a TCP flow or a UDP flow. The specific generation steps are discussed in detail at the end of this section. In the second step, before the feature selection, we first determine the GBDT algorithm as the classifier for DDoS detection and recognition, which is discussed in detail in the fourth part. In the third step, the feature selection method of the random forest feature score and Pearson correlation coefficient is used to select each attack type. In the fourth step, each of the attack vectors are characterized by a feature subset, and an attack vector feature tree is constructed.

DDoS attacks are constantly changing, and each time a DDoS attack occurs, it is often accompanied by multiple attack methods. At present, there are 45 kinds of DDoS attacks based on TCP and UDP protocols [38–40]. Many researchers have proposed DDoS attack classification methods [41–44]. In view of the current types of DDoS attacks, this paper extracts 102 features by summarizing the research results of these researchers [8–11]. As shown in Appendix A Table A1, the features corresponding to sequence numbers 1–102 are based on 102 features of the TCP flow, and the features corresponding to sequence numbers 54–102 are based on 49 features of the UDP flow. These features serve as a collection of original features for detecting DDoS attacks.

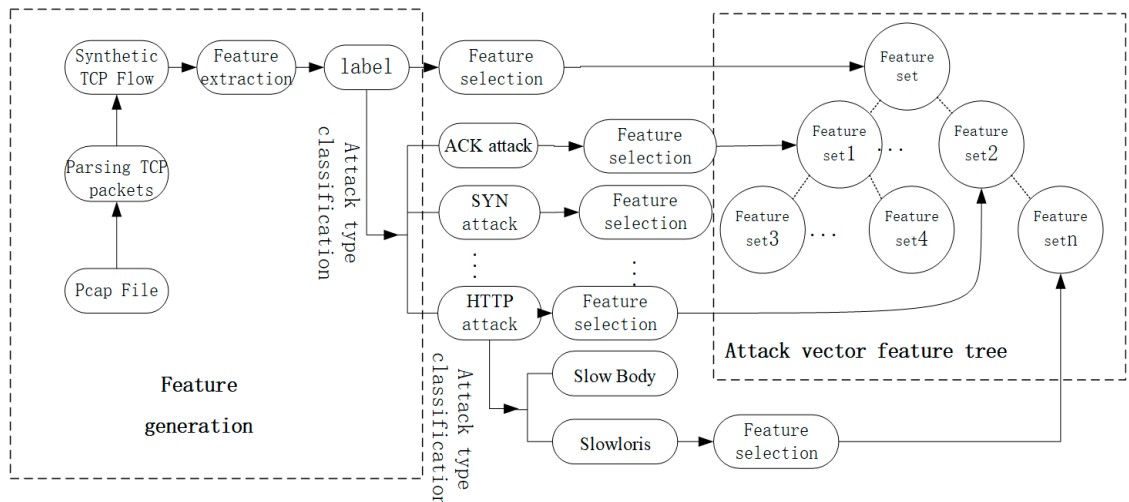

**Figure 1.** Attack vector feature tree generation.

Extracting the important features of the DDoS attack type is very important for early detection of DDoS attacks [11]. A type of DDoS attack tends to highlight several important features. In other words, several features can characterize a DDoS attack. If we can acquire some features, we can lock the DDoS attack type range, which will greatly help the later DDoS mitigation. According to the method proposed by researchers in [41–44], each attack vector can be characterized by feature subsets to construct the attack vector feature tree. When a DDoS attack occurs, the attack feature tree can be used to quickly locate the DDoS attack type.

The goal of constructing a feature tree is to quickly locate which types of attacks appear when a hybrid attack occurs. To clearly describe the construction process of the feature tree, the details are shown in Figure 1 (1) First, the feature of the mixed attack is extracted, and the obtained feature subset is used as the root of the feature tree. (2) Then, the mixed traffic corresponding to the root node is separated into various attack types, and the traffic of each attack type is mixed with the normal traffic. Then, feature extraction is performed, and the obtained feature subset is used as a leaf node (3). Subsequently, iterating through all the leaf nodes is performed and steps (1) and (2) are repeated. Finally, steps (1), (2), and (3) are performed to obtain a feature tree.

According to the attack vector feature tree, as shown in Figure 1, when the feature subset 1 changes, the attack type's range can be quickly locked, that is, the node where the feature subset 1 is located and the DDoS attack type that corresponds to the child node is determined. The attack vector feature tree has the characteristics of a fast perceptual attack and is beneficial to DDoS mitigation. Additionally, it saves time regarding feature selection because machine learning directly uses the changed feature set's learning and prediction.

**Definition 1.** *Feature Set: A collection of features consisting of several features in Appendix A Table A1. A feature set A is represented by [a1, a2, a3, ..., an], n ∈ (1, 2, 3, ..., 102), where n represents the sequence number in Appendix A Table A1, and an represents the feature corresponding to the sequence number n in Appendix A Table A1. For example, a feature set is [1, 6, 7, 9], indicating that the feature set has four features, which are the corresponding features of the sequence numbers in Appendix A Table A1, namely, [syn_in_pps, Push_out_pps, Fin_in_pps, and Rst_in_pps].*

**Definition 2.** *One-way flow: refers to a list of data packets in a TCP or UDP flow that are flowed by the client to the server and arranged in chronological order, or a list of data packets that are sent by the server to the client and arranged in chronological order.*

To quickly extract DDoS features, this paper builds a big data processing framework. The processing flow is shown in Figure 2. Kafka [45] is a message queuing system that can be published and subscribed. Mainly considering the achievability of engineering practice, we, therefore, joined kafka in the experiment. Kafka has a buffering effect, and with kafka the big traffic DDoS attack will not rush our services. Kafka often used to collect real-time data from applications, the data format that Kafka sends to consumers is <key, value, timestamp>. Spark Streaming is a consumer of data. It is an extension of the core Spark API that enables scalable, high-throughput, and fault-tolerant stream processing of real-time data streams. Hbase is a distributed storage database that can store data with different key values at different times. First, the framework parses the fields of each packet of the pcap file into a text file, that is, a packet is parsed into a row of data. Kafka sends the parsed data to SparkStreaming. The data format = <K, V>, where K = sip#dip#sport#dport#protocol is a typical quintuple for reconstructing TCP/UDP flows, V = data packet arrival time, packet size, IPgram, the text version number, the IP header length and other IP packet fields, the TCP packet fields, and the UDP packet fields. SparkStreaming then compares K with K_i in the list to determine whether the stream is normal traffic or attack traffic. List is the label we manually identify (normal, SYN flood, or other), and a quintuple is used to represent a TCP flow or a UDP flow, such as List = [<K1, type1>, <K2, type2> ... <Ki, typei>], where Ki represents the quintuple of a TCP flow or UDP flow i, and i indicates the attack type of a TCP flow or UDP flow i. After comparison, we obtain <K, V, type>, which determines the label of each packet. Finally, it is stored in Hbase in the form of < type, E >, where type refers to the attack type and E refers to the statistical feature set. To calculate <K, V, type> and convert to <type, E>, the specific algorithm is as follows:

(1)    Enter <K, V>.
(2)    Use the map operator in Spark to determine the flow direction of the packet corresponding to <K, V>. It returns <K_in, V> if the packet is flowing from the client to the server; otherwise, it returns <K_out, V>.
(3)    Then, call the reduce operator, and the result returns a one-way flow. For example, return <K_in, (V1, V2, V3, ... , Vn)>.
(4)    The reduce function in (3) calculates the characteristics of Appendix A Table A1. For example, calculate syn_in_pps = countSYN (V1, V2, V3, ... , Vn)/Vn.time-V1.time. countSYN (V1, V2, V3, ... , Vn) is the number of SYN flags in the calculation packet V1, V2, V3, and Vn, which can be calculated according to the flags field of the TCP packet. Vn.time-V1.time indicates that the timestamp of V1 is subtracted from the timestamp of the packet Vn. After all the features are calculated, <K, E> is returned, and E = "e1, e2, e3, ... , en" en represents a certain feature in Appendix A Table A1.
(5)    Then, continue to call the reduce function, merge the features of the two unidirectional flows of the same flow in (4), and return <K, E>.
(6)    Call the JOIN function, List.join(E). E is the result of (5) return. The purpose of calling the JOIN function is to label each E. Call the JOIN function to obtain <K, (type, E)>, return <type, E>.
(7)    Store the results returned by (6) in the Hbase database.

In this paper, the machine learning algorithm can be used to retrieve the normal tag data and some attack type data from the HBase database. This makes it easy to extract the feature set corresponding to a certain attack type.

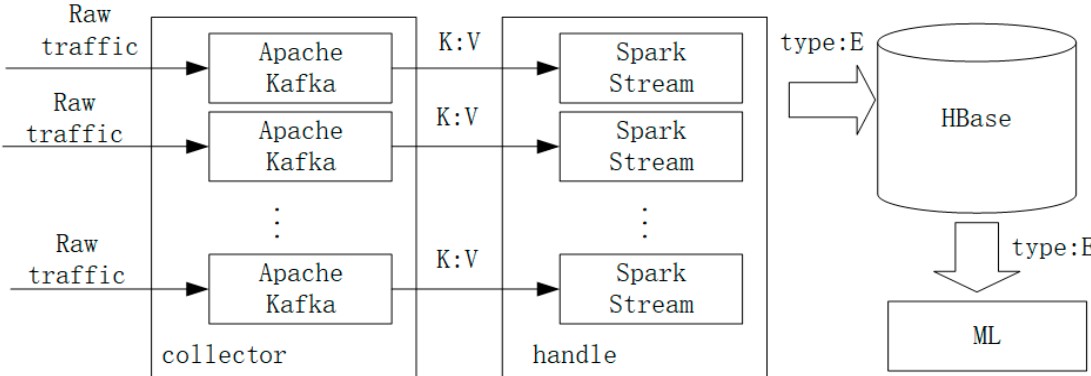

**Figure 2.** Feature processing flow.

## 4. DDoS Classifier

Decision trees are a basic classification and regression method. The decision tree model has a fast classification, but it is also prone to overfitting. When boosting is performed in the classification problem, it learns multiple classifiers by changing the weight of training samples, and linearly combines these classifiers to improve classifier performance. The boosting math is expressed as:

$$f(x) = w_0 + \sum_{m=1}^{M} w_m \varphi_m(x), \tag{1}$$

where $w$ is the weight, $\varphi$ is the set of weak classifiers, $M$ represents the classifiers, and $w_m$ represents the weight of the $m$th classifier. It can be seen that the final classifier is a linear combination of basis functions.

The GBDT algorithm is a boosting algorithm proposed by Friedman [46] in 2001. It is an iterative decision tree algorithm consisting of multiple decision trees, and the conclusions of all trees are added as the final answer. The specific idea is that each time the model is built, the gradient of the model loss function is established in the previous direction, while the traditional boosting idea is to weight the correct and wrong samples. The GBDT algorithm plays two roles in this paper. On the one hand, it implements DDoS classification based on the GBDT algorithm; on the other hand, it uses the GBDT algorithm to evaluate and verify the effect of feature selection.

KNN, SVM, NB, and MLP(Multi-Layer Perceptron) algorithms are algorithms that are often used in intrusion detection algorithms. Here, the paper compares the GBDT algorithm with KNN, SVM, NB, and MLP in the three aspects of accuracy, running time, and the ROC(Receiver Operating Characteristic) curve. The GridSearchCV adjustment parameters are optimized for KNN, SVM, and NB algorithms, but it is difficult for MLP to determine the optimal parameters. It can only adjust a good effect according to personal experience. The results are shown in Figures 3–5, respectively. In the TCP flow, the attack traffic types are salphfl, malphfl, alphfl, mptmp, mptp, ptmp, ntscSYN, sntscSYN, ptmpHTTP, and mptpHTTP. In UDP flows, the attack traffic type refers to ptpposcaUDP. These types of attacks come from the division of the WIDE dataset [47], such as the attack type ntscSYN, which refers to network_scan_SYN. Random extraction of 1 k, 10 k, and 100 k TCP or UDP datasets from the Hbase database is performed. This paper uses GBDT, KNN, SVM, NB, and MLP as the five machine learning algorithms to classify 1 k TCP and 1 k UDP datasets. The results show that when classifying TCP traffic, the training accuracy of the GBDT algorithm reaches 0.99, and the test accuracy rate reaches 0.98. The training accuracy of the KNN algorithm reaches 0.89, and the test accuracy rate is 0.9, while the training accuracy and test accuracy of SVM algorithm are 0.89 and 0.95, respectively. The training accuracy and test accuracy of the MLP algorithm are 0.96 and 0.93, respectively. The lowest is the NB algorithm, and the training scores and test accuracy rates are 0.56 and 0.6, respectively. It can be seen that the accuracy of the GBDT algorithm is significantly better than that of the other algorithms when the dataset's size

is 1 k. As the amount of data increases, the accuracy of the GBDT, KNN, SVM, and MLP algorithms decreases, but the accuracy of the NB algorithm barely fluctuates, and the accuracy is poor. The NB algorithm is suitable for datasets with few features, and eigenvalues that are discrete values and few in number. The features extracted in this paper are continuous values and have 102 features, which is obviously not suitable for NB algorithm. When the dataset's size is 10 k, the training accuracy and test accuracy of the GBDT algorithm are reduced, which are 0.959 and 0.962, respectively. The accuracy of the test is higher than that of the KNN, SVM, and MLP algorithms. Similarly, when the amount of data is increased to 100 k, the test accuracy of the GBDT algorithm is higher than that of the KNN, SVM, and MLP algorithms. For UDP flows, when the dataset's size is 1 k, the training accuracy and test accuracy of the GBDT algorithm are 0.994 and 0.95, respectively. Regardless of the training accuracy or the test accuracy, the GBDT algorithm achieves better results than other algorithms. As the size of the datasets increases, the accuracy of all algorithms decreases. However, the performance of the GBDT algorithm is superior to other algorithms. The reason for the decrease in accuracy may be due to the inaccurate direction of the flow during the feature extraction phase. In general, the training accuracy and test accuracy of TCP are higher than the training accuracy and test accuracy of UDP in the corresponding algorithm because TCP traffic has a richer DDoS attack signature.

In terms of training time, as the amount of data increases, the training time of the GBDT algorithm also increases, and the training time of the GDBT is not optimal in the five algorithms. However, when the data volume is 100 k, the GBDT test time on the TCP datasets is only 0.031 s, and the test time on the UDP datasets is only 0.023 s, which is smaller than that of the KNN, SVM, and MLP algorithms.

As shown in Figure 4, the training time and test time of the SVM algorithm are very long. When the dataset's size is 100 k TCP datasets, the time used is 451 and 25.22 s, respectively. When the dataset's size is 100 k UDP datasets, the time used is 202 and 9.973 s, respectively. The MLP algorithm also takes a long time. The training time and test time are 28.456 and 1.02 s, respectively, compared to the 100 k TCP datasets. Although the training time of the KNN algorithm is relatively short, the test time is relatively long. The test times for the 100 k TCP datasets and the 100 k UDP datasets are 47.77 and 6.925 s, respectively. The test time is very important for DDoS detection. The shorter the test time, the earlier it is possible to detect whether a DDoS attack has occurred, thus preventing the DDoS attack from posing a greater threat to the server. Obviously, the GBDT algorithm is superior to the other algorithms in terms of accuracy and test time performance. The comparison of the ROC curves of the GBDT algorithm with other algorithms is shown in Figure 5. The ROC curve of the GBDT algorithm encloses the ROC curve of the other algorithms. In summary, the time to detect DDoS attacks with the GBDT algorithm is shorter than that of the other algorithms, and the accuracy of detecting DDoS attacks is higher than that of the other algorithms. Therefore, the GBDT algorithm is selected as the DDoS attack detection classifier.

For the TCP flow, this paper simply collects 102 features using the GBDT algorithm without optimization to achieve a test accuracy of approximately 95%; for UDP flows with a total of 49 features, the GBDT algorithm achieves a test accuracy of 91%. The selected features of Appendix A Table A1 and the GBDT algorithm are valid for DDoS attack traffic detection. However, due to the large number of features, feature selection is required to improve the generalization ability of the algorithm and shorten the detection time.

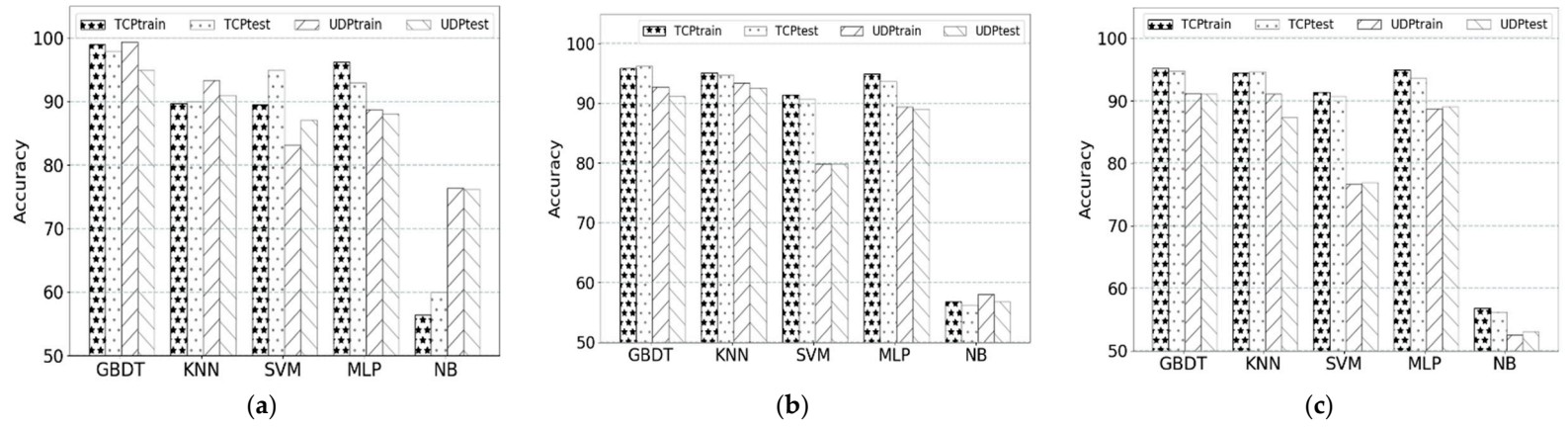

**Figure 3.** Accuracy comparison. (**a**) 1 k data; (**b**) 10 k data; (**c**) 100 k data.

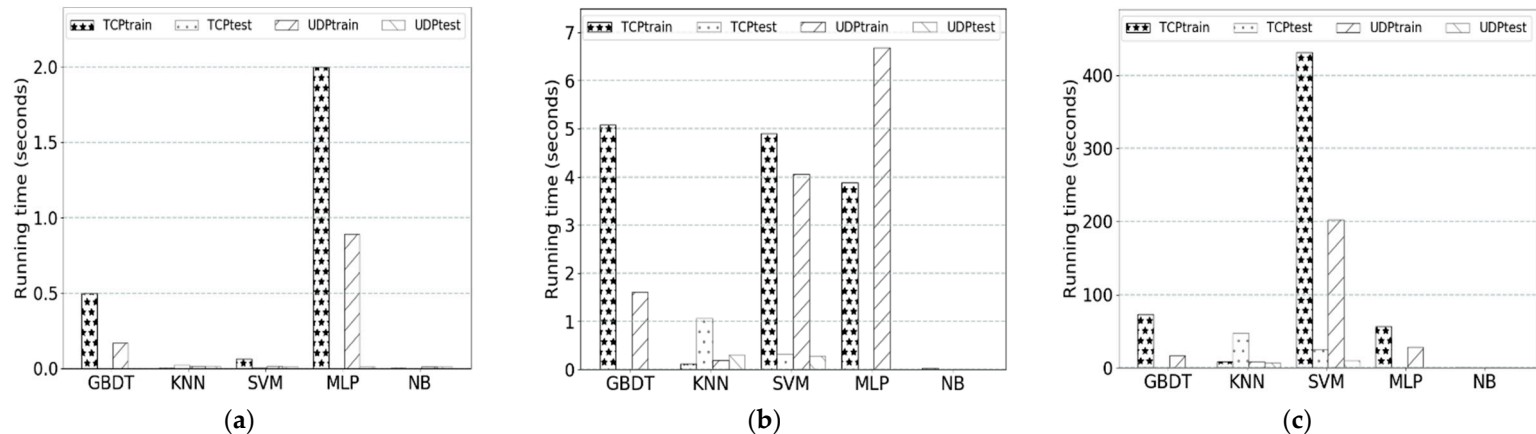

**Figure 4.** Running time comparison. (**a**) 1 k data; (**b**) 10 k data; (**c**) 100 k data.

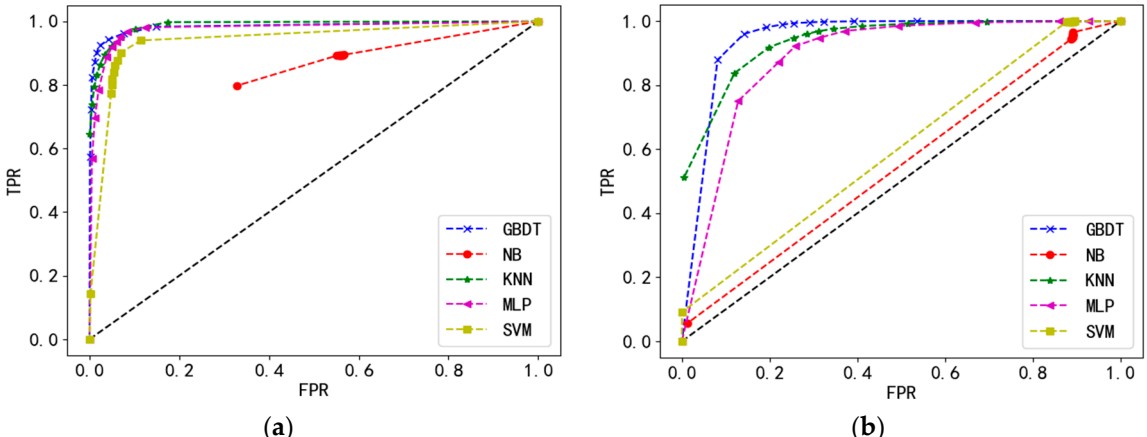

**Figure 5.** ROC curve comparison. (**a**) ROC comparison of TCP flow datasets; (**b**) ROC comparison of UDP flow datasets.

## 5. DDoS Classifier Optimization

The previous part has shown that the GBDT algorithm as a DDoS attack classifier is better than other algorithms. This part further considers the GBDT classifier optimization from two aspects. The first is feature selection, and the second is the specific classifier tuning.

### 5.1. Feature Selection

Feature selection has two purposes. First, for each DDoS attack, several features that best characterize the attack behavior are selected. Second, feature selection can help improve detection accuracy, reduce the false positive rate, and thus accurately use the classifier. Choosing a good attack feature can not only effectively identify DDoS attack traffic but also improve the response speed of the algorithm if the recognition rate allows.

Feature selection methods are divided into three types: filtering methods, packaging methods, and embedding methods. The main idea of the filtering method is to score the importance of each feature and then select the features according to the ranking of the scores. The main methods are chi-squared test, information gain, and correlation coefficient. The main idea of the packaging method is to regard the selection of the subset as a search optimization problem, generate different combinations, evaluate the combination, and compare it with other combinations. Therefore, this approach regards the selection of subsets as an optimization problem. The main method is the recursive feature elimination algorithm. The embedding method identifies the attributes that are important to the model training during the process of determining the model. Feature selection can also be combined with the artificial ant colony algorithm, artificial bee colony algorithm, genetic algorithm, annealing algorithm, and other algorithms to obtain the best features. This paper proposes a new integrated method of feature selection based on the random forest and Pearson correlation coefficients.

Random forests are a common feature selection method. In terms of DDoS attack feature selection, Robin. G et al. [48] proposed a feature importance index based on random forests, using random forest importance scores to gradually increase features. Zahangir Alam et al. [49] used a random forest importance score to rank and extract top-ranking features. Random forests can effectively extract the importance scores of features but cannot distinguish the correlation between features.

The method in this paper aims to calculate and rank all feature importance scores using random forests. The feature whose feature importance score is less than $\Omega$ is removed, and $\Omega$ is a threshold for calculating the feature importance score using the random forest. The rationale of removing the importance scores that are less than $\Omega$ is mainly to remove those features that are not related to the classification category or those that are extremely related to the classification category. This paper

introduces the Pearson correlation coefficient, which examines the degree of correlation between two things. If there are two characteristics, *X* and *Y*, their correlation is calculated as follows:

$$\rho_{x,y} = \frac{N \sum XY - \sum X \sum Y}{\sqrt{N \sum X^2 - (\sum X)^2} \sqrt{N \sum Y^2 - (\sum Y)^2}}, \tag{2}$$

where *N* is the size of the record. The meaning of the finally calculated correlation coefficient can be understood as follows: (1) When the correlation coefficient is 0, the *X* and *Y* characteristics have no relationship. (2) When the value of *X* increases (decreases) and the value of *Y* increases (decreases), the two features are positively correlated, and the correlation coefficient is between 0.00 and 1.00. (3) When the value of *X* increases (decreases) and the value of *Y* decreases (increases), the two features are negatively correlated, and the correlation coefficient is between −1.00 and 0.00. The larger the absolute value of the correlation coefficient is, the stronger the correlation, and the closer the correlation coefficient is to 1 or −1, the stronger the correlation, while the closer the correlation coefficient is to 0, the weaker the correlation.

---

**Algorithm 1: Feature selection algorithm**

---

**Input:** datasets D;Feature set A;Learning algorithm GBDT;
Threshold value of feature importance score $\Omega$; Pearson correlation coefficient threshold $\rho_0$
**Output:** feature subset A2

1:  Calculate the importance score of feature set A using a random forest.
2:  Sort the scores and select the feature subset A1 with a score greater than $\Omega$
3:  for i in |A1|:
4:      for j in |A1|:
5:          $\varrho$ = Pearson(i,j)
6:          if $\varrho > \rho_0$ then:
7:              list.add([i,j])
8:  Combine related features in the list to obtain a new list 1
9:  for h in list1:
10:         A1 = A1-lowerscore(h)
11: Test = 0, Set A2 = {}
12: A1 sorts in descending order of score
13: for k in A1:
14:         testscore = CroossGBDT(A2, k)
15:         if testscore > test
16:             A2.add(k), test = testscore
17: Output feature subset A2

---

This paper proposes a feature selection method based on a random forest and Pearson correlation coefficients. The random forest is used to calculate the importance score for each feature. The features with a feature importance score that is higher than $\Omega$ are then combined into a feature subset. The pairwise features of the resulting feature subset are calculated for their Pearson correlation coefficients. The feature for which the Pearson correlation coefficient $\rho_{x,y} > \varrho 0$ ($\varrho 0$, the Pearson correlation coefficient threshold) has the smallest importance score is removed. The remaining feature subsets are sequentially input into the GBDT algorithm. If the feature and the previous feature together improve the test accuracy of the GBDT algorithm, the feature is selected as the result feature; otherwise, the feature is not added as the result feature. The specific algorithm is shown in Algorithm 1. The Pearson(i, j) function of the fifth line of the algorithm refers to the Pearson correlation coefficient for calculating the feature i and the feature j. Line 7 describes that if the Pearson correlation coefficient of feature i and feature j is greater than $\varrho 0$, then features i and j are added to the list. In the algorithm,

list refers to a list of feature sets, for example list = [[1, 2], [4, 5], [5, 13]], where the Pearson correlation coefficient of features 41 and 42 is greater than $\varrho 0$, so feature 41 and 42 join the list, resulting in a new list given by list = [[1, 2], [4, 5], [5, 13], [41, 42]]. Line 8 refers to the combination of two related features, such as combining the two related features in the list to obtain list = [[1, 2], [4, 5, 13], [41, 42]]. The function of lowerscore(h) on line 10 indicates that features with lower importance scores are obtained. For example, when h = [4, 5, 13], feature 4 scores the highest, so the function lowerscore(h) returns the set [5, 13]. The 14th line shows the function CroossGBDT(A2, k), which refers to a new feature subset consisting of a feature subset and a feature k, and the new feature subset is input into the GBDT algorithm to return the test set. The feature selection algorithm in this paper is called RFPW (random forest and Pearson wrap).

### 5.2. GBDT Algorithm Parameters

After the feature selection, in order to obtain higher accuracy, it is necessary to adjust the parameters of the GBDT algorithm. The optimization of the GBDT algorithm's parameters ensures that the GBDT algorithm achieves better detection accuracy and better generalization ability. There are four main parameters of the GBDT algorithm, namely, the number of the largest weak learner (CART tree), the subsampling, the condition of the subtree's continual division, and the maximum depth of the subtree. The number of the largest weak learners is represented by n_estimators, such that when n_estimators are too small, it is easy to underfit; in contrast, when n_estimators are too large, it is easy to overfit. The sampling uses the subsample to indicate that the GBDT algorithm does not put back the sample. If the value is less than 1, only a part of the sample will be used for the decision tree fitting of GBDT. The condition in which the tree continues to be divided is represented by min_samples_split, which limits the conditions under which the subtree continues to be partitioned. The maximum depth of the tree is represented by max_depth, which can easily cause overfitting. These four parameters are important for creating a robust GBDT model.

This article starts with the number of weak learners (n_estimators). The result is shown in Figure 6. When n_estimators = 400, the training accuracy is optimal. The accuracy of the number of iterations drops rapidly after 400. Figure 7 shows that the maximum depth max_depth is not as large as possible, and the minimum division set min_samples_split is not as small as possible. When the maximum depth max_depth is small, the effect decreases as the minimum division set increases. When max_depth = 16, min_samples_split = 150, and the precision is 0.9997. Figure 8 shows that the subsample value 0.25 is optimal. This paper uses the parameters n_estimators = 400, max_depth = 16, min_samples_split = 150, subsample = 0.25 as the optimized model.

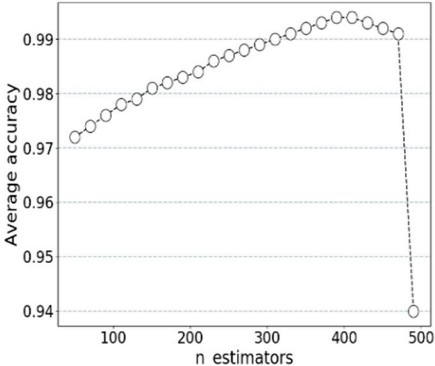

**Figure 6.** n_estimators and accuracy.

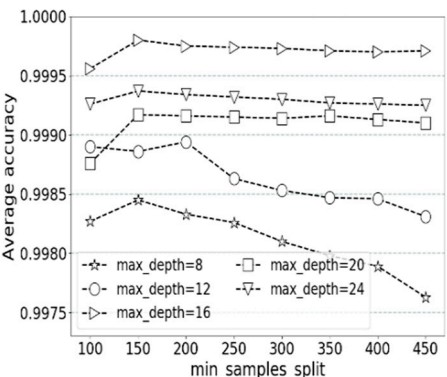

**Figure 7.** max_depth and min_samples_split.

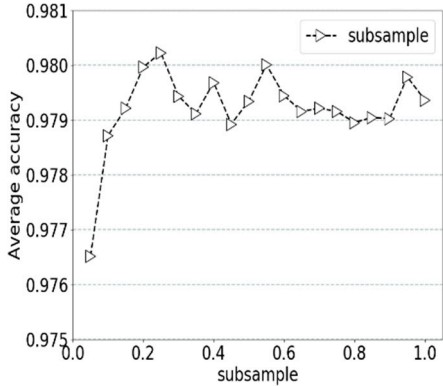

**Figure 8.** Subsample and accuracy.

## 6. The Experimental Results and Analysis

### 6.1. Datasets

This paper uses the MAWI (Measurement and Analysis on the WIDE Internet) datasets. The datasets consist of packet tracking from MAWI archives and was released publicly in 2016. Each trace in this database is a pcap file containing traffic captured within 15 min of a specific date since 2001, captured on a trans-Pacific link between Japan and the United States. This paper uses the 201710281400.pcap data package [47] collected at 2 p.m. on October 28, 2017. The MAWI datasets is divided into 10 categories [50]. The types of attack traffic used in this experiment are shown in Table 2.

**Table 2.** 201710281400.pcap datasets tag list.

| Types of Attack | Specific Type |
|---|---|
| Normal | |
| Alpha flow attack | salphfl |
| | malphfl |
| | alphfl |
| Multi. points attack | mptmp |
| | mptp |
| | ptmp |
| SYN attack | ntscSYN |
| | sntscSYN |
| HTTP attack | ptmpHTTP |
| | mptpHTTP |
| UDP attack | ptpposcaUDP |

The experimental platform configuration is as follows: Intel(R) core(TM) i5-8250U CPU @ 1.6 GHz-1.8 GHz, memory (RAM) 8 G, 64-bit operating system, Windows 10. In this paper, the machine learning algorithm is used to call the algorithm provided by the sklearn library.

*6.2. Performance Measurement*

The performance of an anomalous intrusion detection system is assessed by its ability to properly classify events as attacks or normal behavior. Based on the true nature of the given event and the prediction of IDS, the four possible outcomes can be understood by the confusion matrix given in Table 3 [11]. Various indicators have been used for performance evaluation. Some key indicators include accuracy, accuracy rate, false positive rate, F measurement, and recall [51]. True positive (TP): an event that is actually an attack and successfully marked as an attack; true negative (TN): an event that is actually normal and successfully marked as normal; false positive (FP): a normal event classified as an attack; false negative (FN): an attack that is incorrectly classified as a normal event.

**Table 3.** Binary classification confusion matrix.

| Subject | | Predictive Output | |
|---|---|---|---|
| | | **1** | **0** |
| **Actual Output** | **1** | TP | FP |
| | **0** | FN | TN |

The accuracy formula is as follows:

$$ACC = \frac{TP + TN}{TP + TN + FP + FN}. \tag{3}$$

The detection rate formula is as follows:

$$TPR = \frac{TP}{TP + FN}. \tag{4}$$

The false positive rate formula is as follows:

$$FPR = \frac{FP}{FP + TN}. \tag{5}$$

The precision rate formula is as follows:

$$PR = \frac{TP}{TP + FP}. \tag{6}$$

The recall rate formula is as follows:

$$RR = \frac{TP}{TP + FN}. \tag{7}$$

The F_1 measurement formula is as follows:

$$F_1 = \frac{2*PR*RR}{PR + RR}. \tag{8}$$

*6.3. Normalization*

The data are normalized prior to algorithm input, and the processing formula is as follows:

$$\acute{x} = \frac{x - min}{max - min}, \tag{9}$$

where $x$ represents any value in an attribute in a record, min is the minimum value of the attribute, and max is the maximum value of the attribute.

### 6.4. Cross-Validation I

The purpose of cross-validation is to obtain a reliable and stable model, such that K-fold cross-validation divides the datasets into K subsamples, where a single subsample is retained as the data of the verification model, and the other K-1 samples are used for training. The cross-validation is repeated K times, each subsample is verified once, and the average results of K-times are used as the final result. This article uses a 10-fold cross-validation technique.

### 6.5. Feature Selection Algorithm Experiment

6.5.1. Determine the $\varrho 0$ Value

$\varrho 0$ is a parameter of the RFPW algorithm that represents the value of the Pearson coefficient correlation. When the $\varrho 0$ values are different, the results obtained by the RFPW algorithm are also different. When the value of $\varrho 0$ is small, it means that as long as the feature has a slight correlation, the feature with the small importance score is removed, which will delete many features and even remove some useful features. When the value of $\varrho 0$ is large, it means that the feature correlation is very large and then the feature with the smaller importance score is removed, so that fewer features are removed. However, feature redundancy can also be introduced, which reduces the performance of the algorithm. Therefore, the value of $\varrho 0$ determines the quality of the RFPW algorithm. The experimental results (Tables 4 and 5) show the characteristics obtained by the RFPW algorithm for different values of $\varrho 0$. For TCP traffic, when $\varrho 0$ is greater than or equal to 0.6, the feature tends to be stable, for a total of 10 features. For UDP traffic, when $\varrho 0$ is greater than or equal to 0.3, the feature tends to be stable, for a total of six features. The accuracy ratio corresponding to the datasets after feature selection that is input to the GBDT algorithm is shown in Figure 9. For TCP traffic, as $\varrho 0$ becomes larger, the accuracy increases, but when $\varrho 0$ is greater than or equal to 0.6, the accuracy fluctuates at 0.95. For UDP traffic, as $\varrho 0$ becomes larger, the accuracy increases, but when $\varrho 0$ is greater than or equal to 0.3, the accuracy fluctuates at 0.92. Therefore, when detecting the TCP flow, $\varrho 0 = 0.6$ is used in the RFPW algorithm. When detecting the UDP flow, $\varrho 0 = 0.3$ is used in the RFPW algorithm.

**Table 4.** Features acquired on the TCP flow dataset when $\varrho 0$ values are different.

| $\varrho_0$ | Feature Number | Feature Subset |
|---|---|---|
| 0.1 | 2 | [27, 76] |
| 0.2 | 2 | [27, 76] |
| 0.3 | 6 | [9, 10, 27, 33, 76, 80] |
| 0.4 | 6 | [9, 10, 27, 33, 76, 80] |
| 0.5 | 6 | [9, 10, 27, 33, 76, 80] |
| 0.6 | 6 | [9, 10, 27, 33, 76, 80] |
| 0.7 | 6 | [9, 10, 27, 33, 76, 80] |
| 0.8 | 6 | [9, 10, 27, 33, 76, 80] |
| 0.9 | 6 | [9, 10, 27, 33, 76, 80] |

**Table 5.** Features acquired on the UDP flow dataset when $\varrho 0$ values are different.

| $\varrho_0$ | Feature Number | Feature Subset |
|---|---|---|
| 0.1 | 2 | [27, 76] |
| 0.2 | 2 | [27, 76] |
| 0.3 | 6 | [9, 10, 27, 33, 76, 80] |
| 0.4 | 6 | [9, 10, 27, 33, 76, 80] |
| 0.5 | 6 | [9, 10, 27, 33, 76, 80] |
| 0.6 | 6 | [9, 10, 27, 33, 76, 80] |
| 0.7 | 6 | [9, 10, 27, 33, 76, 80] |
| 0.8 | 6 | [9, 10, 27, 33, 76, 80] |
| 0.9 | 6 | [9, 10, 27, 33, 76, 80] |

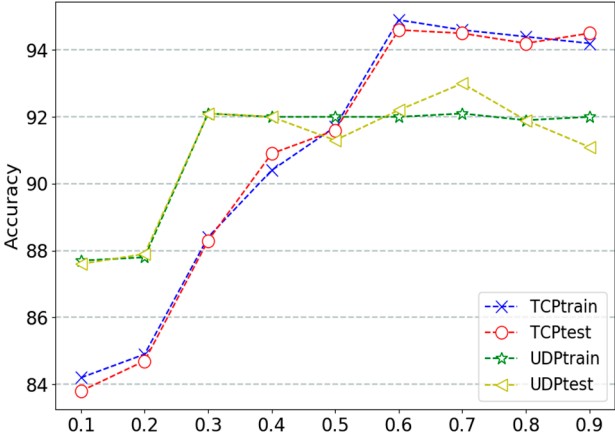

**Figure 9.** The abscissa is the value of $\varrho 0$.

6.5.2. Comparison before and after Feature Selection

In this paper, the 20 k TCP flow dataset and the 20 K UDP flow dataset are used as feature selection before and after comparison. First, for the TCP datasets, a feature subset of $\varrho 0 = 0.6$ is selected as the input to the GBDT algorithm. For the UDP datasets, a feature subset of $\varrho 0 = 0.3$ is selected as the input to the GBDT algorithm. The accuracy before and after feature selection is shown in Figure 10. Whether it is a TCP flow dataset or a UDP dataset, it is improved accurately after feature selection.

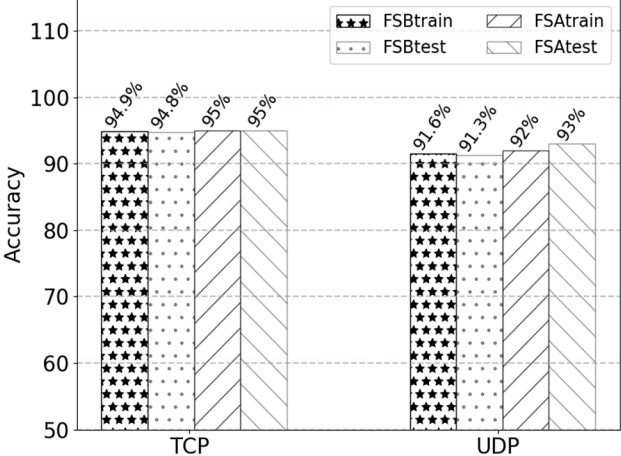

**Figure 10.** FSBtrain represents the training accuracy before feature selection, and FSBtest represents the test accuracy before feature selection. FSAtrain indicates the training accuracy rate after feature selection, and FSAtest indicates the test accuracy rate after feature selection.

6.5.3. Comparison of RFPW and the Dimensionality Reduction Algorithm

In this paper, the 20 k TCP flow datasets and the 20K UDP flow datasets are used as the RFPW algorithm to compare the three dimensionality reduction algorithms of PCA, SVD, and LDA. First, for the TCP datasets, a feature subset of $\varrho 0 = 0.6$ is selected as the input to the GBDT algorithm. For the UDP datasets, a feature subset of $\varrho 0 = 0.3$ is selected as the input to the GBDT algorithm. The accuracy and number of features before and after feature selection are shown in Figures 11 and 12.

Whether it is a TCP flow dataset or a UDP dataset, the RFPW algorithm is superior to the dimensionality reduction algorithm in accuracy. For the TCP flow dataset used with RFPW, as long as 10-feature accuracy is selected, it will reach approximately 95%, while the PCA (principal component analysis) algorithm and the SVD (singular value decomposition) algorithm need to retain 30 features (dimensions) or more to reach 92%. As there are only two classifications in our experiment, the LDA (linear discriminant analysis) algorithm is directly reduced to one dimension. However, the accuracy

rate is very large, only approximately 88%. For UDP datasets, the RFPW algorithm can achieve an accuracy of 93% with only six features, while the SVD algorithm requires 11 features to achieve an 88% accuracy. PCA requires 11 features to achieve 89% accuracy. LDA has an accuracy of only approximately 75%.

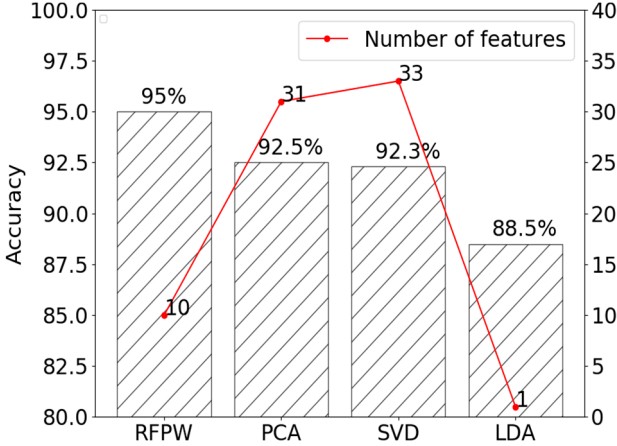

**Figure 11.** TCP flow datasets feature selection and dimensionality reduction.

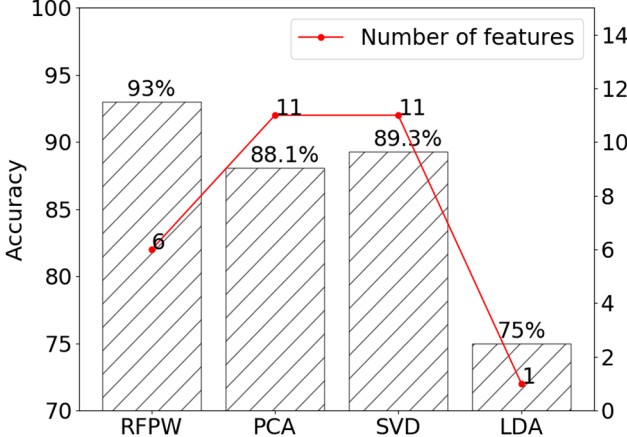

**Figure 12.** UDP flow datasets feature selection and dimensionality reduction.

6.5.4. Feature Selection Comparison

Jarrah et al. [18] proposed random forest-forward selection sorting (RF-FSR) and random forest-backward sorting (RF-BER). The experimental results show that the selected features on the KDD-Cup 99 dataset effectively improve their detection rate and reduce the false positive rate, reaching 99.8% and 0.001%, respectively. Wei W et al. [15] used information gain and chi-square methods to rank the importance of the 41 attributes extracted from network traffic. The Bayesian network and C4.5 algorithm are then used to detect the attack and determine the size of the attribute suitable for fast detection. The empirical results show that using only the most important nine attributes, the detection accuracy remains the same or even achieves some improvement compared to all 41 attributes based on the Bayesian network and the C4.5 method. O.Y. Al Jarrah et al. [19] proposed combining the consistent subset evaluation (CSE) and DDoS characteristic features (DCF) techniques in feature selection algorithms to identify and select the most important and relevant features associated with DDoS attacks. This paper compares the RF-FSR algorithm [18], RF-BER algorithm [18], GI (information gain) [15], CS (chi-square test) [15], and CSE [19]. In this paper, different methods are used to select features using the 20k TCP flow dataset and 20k UDP flow dataset. The selected feature subset results are shown in Tables 6 and 7. RF-FSR, RF-BER, GI, and CS are feature selection algorithms, and the

selected feature subsets are large in size. The accuracy of the features selected by different algorithms is shown in Figure 13.

**Table 6.** Comparison of TCP flow dataset feature selection algorithms.

| Method | Feature Subset Number | Feature Subset |
|:---:|:---:|:---|
| RF-FSR | 18 | [6, 13, 18, 27, 38, 39, 47, 50, 51, 56, 57, 60, 73, 76, 82, 84, 88, 92] |
| RF-BER | 24 | [6, 13, 18, 27, 38, 39, 47, 50, 51, 56, 57, 60, 68, 73, 76, 80, 82, 84, 85, 88, 89, 92, 98, 101] |
| GI | 35 | [1, 12, 13, 14, 39, 43, 44, 45, 50, 51, 59, 60, 63, 64, 65, 66, 67, 71, 72, 73, 80, 81, 82, 83, 84, 85, 87, 88, 89, 90, 91, 92, 93, 94, 95] |
| CS | 34 | [13, 14, 15, 16, 17, 18, 21, 27, 39, 45, 46, 60, 61, 64, 65, 66, 67, 68, 69, 72, 74, 80, 81, 88, 89, 90, 93, 94, 95, 96, 99, 100, 101, 102] |
| CSE | 16 | [6, 13, 18, 27, 38, 39, 47, 50, 51, 57, 60, 76, 82, 84, 88, 92] |

**Table 7.** Comparison of feature selection algorithms for UDP flow datasets.

| Method | Feature Subset Number | Feature Subset |
|:---:|:---:|:---|
| RF-FSR | 20 | [0, 1, 34, 35, 36, 37, 38, 39, 42, 43, 44, 45, 46, 47, 16, 17, 18, 19, 20, 26] |
| RF-BER | 16 | [0, 1, 16, 17, 18, 19, 20, 26, 35, 37, 39, 42, 43, 44, 45, 47] |
| CS | 11 | [37, 16, 17,18, 19, 26, 27,42, 44, 45,1] |
| GI | 10 | [37, 16, 18, 19, 26, 27,42, 43, 44, 45] |
| CSE | 7 | [37, 18, 19, 26, 27,42, 44] |

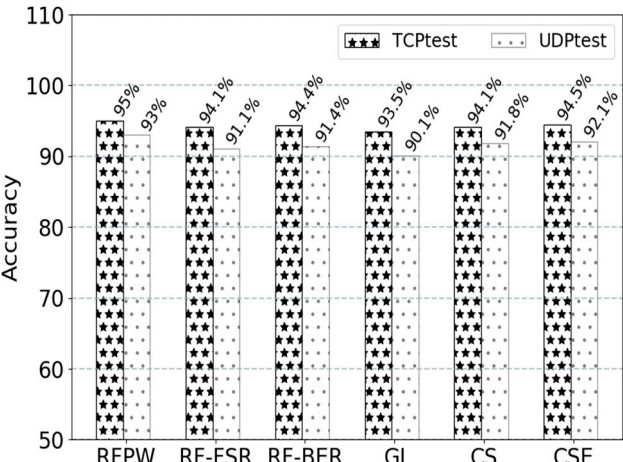

**Figure 13.** TCPtest and UDPtest represent the accuracy of the test set in the TCP flow and UDP flow data, respectively.

### 6.5.5. Comparison of Three Models

Tables 8 and 9 compare the performance indicators of the three models of the GBDT algorithm. The three models are the unoptimized GBDT model, the GBDT model with feature selection, and the model of feature selection and parameter optimization. For the convenience of description, we recorded the three models as GBDT1, GBDT2, and GBDT3. The results show that the optimized GBDT algorithm has very high accuracy and a very low false detection rate. Table 8 shows the performance comparison of the three models under the 100 k TCP flow datasets. Table 9 shows the performance comparison of the three models under the 100 k UDP flow datasets.

**Table 8.** Performance comparison of three models of 100 k WIDE TCP flow datasets.

| Model | ACC | TPR | FPR | PR | RR | $F_1$ |
|-------|-----|-----|-----|----|----|----|
| GBDT1 | 0.9548 | 0.9784 | 0.1499 | 0.9365 | 0.9784 | 0.9569 |
| GBDT2 | 0.9516 | 0.9801 | 0.1455 | 0.9555 | 0.9803 | 0.9677 |
| GBDT3 | 0.9997 | 0.9997 | 0.0007 | 0.9997 | 0.9997 | 0.9997 |

**Table 9.** Performance comparison of three models of 100 k WIDE UDP flow datasets.

| Model | ACC | TPR | FPR | PR | RR | $F_1$ |
|-------|-----|-----|-----|----|----|----|
| GBDT1 | 0.9221 | 0.8706 | 0.0270 | 0.9695 | 0.8706 | 0.9174 |
| GBDT2 | 0.9214 | 0.8598 | 0.0256 | 0.9666 | 0.8598 | 0.9101 |
| GBDT3 | 1.0000 | 0.9999 | 0.00003 | 1.0000 | 0.9999 | 0.9999 |

### 6.5.6. DDOS Attack Vector Feature Tree

This experiment extracts each dataset based on a TCP protocol attack and mixes it with the normal datasets and then selects the feature of these mixed datasets. The goal is to obtain the feature subset corresponding to each attack type. As shown in Figure 14, the attack range can be quickly locked according to the change of the feature. If a certain attack feature set indicates that the leaf node is not a known leaf node but is a child node of a certain node A, then it can be determined that the attack is a node A. Corresponding to the variant of the attack type, decision support is provided to the decision maker.

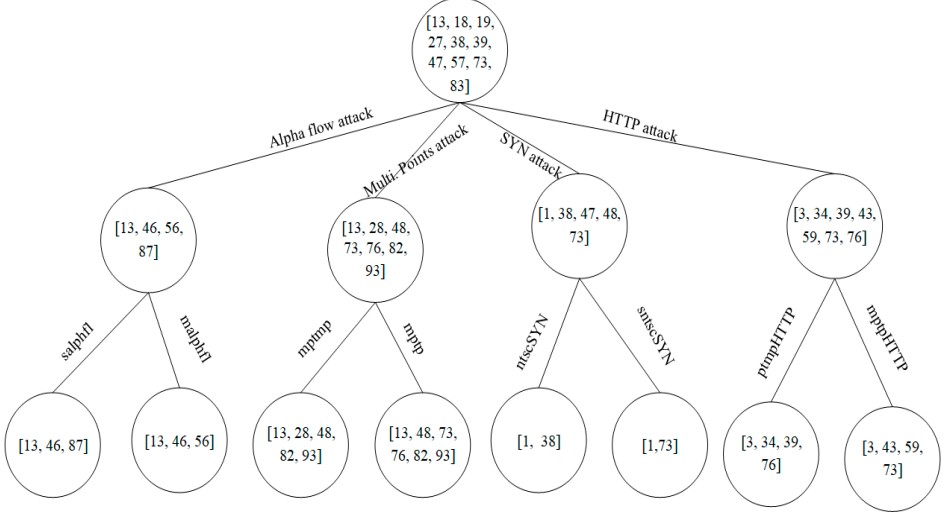

**Figure 14.** Attack vector feature tree.

## 7. Conclusions

This paper first used 102 features to detect DDoS attacks based on TCP protocol and 49 features to detect DDoS attacks based on UDP protocol. In this paper, the GBDT algorithm was compared with the following four algorithms in terms of detection accuracy and running time: KNN, NB, SVM and MLP. The results show that the detection accuracy using the GBDT algorithm is higher than the other algorithms, and the test time is shorter than other algorithms. The RFPW feature selection algorithm combines random forest scores with Pearson correlation coefficients as a search strategy, using the GBDT algorithm as the evaluation criteria. This paper compared RFPW with the traditional dimensionality reduction algorithms and feature selection algorithms. Experiments show that the number of features selected by the RFPW algorithm is small. As long as the DDoS hybrid attack based on the TCP protocol has only 10 features and the DDoS hybrid attack of the UDP protocol requires

only six features, the detection accuracy remains the same or increases. At the end of this paper, the parameters of the GBDT algorithm were tuned, so that the accuracy of DDoS attack detection based on the TCP protocol reaches 0.9997, and the false positive rate is only 0.0007. The detection rate of DDoS attacks based on the UDP protocol reached 100%, and the false detection rate was only 0.00003. The experimental results show that with the detection features and feature selection algorithms proposed in this paper, the GBDT detection classifier and GBDT parameter tuning can provide fast and accurate detection for DDoS attacks, which is meaningful for DDoS mitigation. At the end of the article, the attack special certificate tree was given, and the attack range was quickly locked according to the attack characteristics. DDoS hybrid attack detection will greatly help DDoS mitigation. In reality, it is based on mixed DDoS attacks. Constructing a DDoS attack tree with a small number of different traffic characteristics is useful for quickly locating DDoS attack types and then issuing policies or switching mitigation DDoS models.

**Author Contributions:** J.Z. proposed to amend the comments and deepen the theoretical part of the thesis. Q.L. designed the main method of the thesis and did all the experiments. R.J. helped us to modify the small errors and experimental data collection of the paper. X.L. helped us review the paper.

**Funding:** This research was funded by Research on Theory and Method of Intelligent Monitoring of Service State of High Speed Railway Infrastructure Based on Machine Vision, grant number U1734208. And the APC was funded by National Natural Science Foundation of China Youth Science Fund Project, Flexible Distribution and Scheduling of Memory in Cloud Computing Environment, grant number 61602523.

**Conflicts of Interest:** The authors declare no conflict of interest

## Appendix A

**Table A1.** Distributed denial of service (DDoS) attack characteristics.

| No. | Feature | Description |
| --- | --- | --- |
| 1 | syn_in_pps | SYNACK-tagged TCP packet flowing out every second |
| 2 | synack_out_pps | Streaming TCP packets containing ACK tags per second |
| 3 | Ack_in_pps | Outgoing ACK tag TCP packet per second |
| 4 | Ack_out_pps | Flow into the TCP packet containing the PUSH tag every second |
| 5 | Push_in_pps | Outgoing packet containing the PUSH tag TCP every second |
| 6 | Push_out_pps | FIN tagged TCP packet flowing in per second |
| 7 | Fin_in_pps | FIN tagged TCP packet flowing out per second |
| 8 | Fin_out_pps | Streaming TCP packets containing RST tags per second |
| 9 | Rst_in_pps | Outgoing UDP tagged TCP packet per second |
| 10 | Rst_out_pps | Flowing into unmarked TCP packets per second |
| 11 | Other_in_pps | SYNACK-tagged TCP packet flowing out every second |
| 12 | Syn_in_pps/In_pps | |
| 13 | Syn_in_pps/(Syn_in_pps + Synack_out_pps) | |
| 14 | Syn_in_pps/(Syn_in_pps + Ack_in_pps) | |
| 15 | Ack_in_pps/in_pps | |
| 16 | Ack_in_pps/(Ack_out_pps + Ack_in_pps) | |
| 17 | Ack_in_pps/(Rst_out_pps + Ack_in_pps) | |
| 18 | Push_in_pps/In_pps | |
| 19 | Push_in_pps/(Push_in_pps + Push_out_pps) | |
| 20 | Push_in_pps/(Push_in_pps + Rst_out_pps) | |

**Table A1.** *Cont.*

| No. | Feature | Description |
|---|---|---|
| 21 | Push_in_pps/(Push_in_pps + ack_out_pps) | |
| 22 | Rst_in_pps/in_pps | |
| 23 | Rst_Out_pps/Out_pps | |
| 24 | Fin_in_pps/in_pps | |
| 25 | Fin_in_pps/(Fin_in_pps + Fin_out_pps) | |
| 26 | other_in_pps/in_pps | |
| 27 | shakehds_pps | Handshake times |
| 28 | crw_in_pps | Streaming TCP packets containing crw tags per second |
| 29 | crw_out_pps | Flow out of the TCP packet containing the crw tag per second |
| 30 | ecn_in_pps | Streaming packets containing ecn tag TCP per second |
| 31 | ecn_out_pps | Ecn tagged TCP packet per second |
| 32 | urg_in_pps | Flowing into the TCP packet containing the urg tag every second |
| 33 | urg_out_pps | Outgoing urg tag TCP packet per second |
| 34 | crw_in_pps/in_pps | |
| 35 | crw_in_pps/Ack_in_pps + crw_in_pps | |
| 36 | ecn_in_pps/in_pps | |
| 37 | urg_in_pps/in_pps | |
| 38 | winsize_in_mean | Window average flowing into the packet |
| 39 | winsize_out_mean | Window average of outgoing packets |
| 40 | num_urgent | Contains the quantity of urgent |
| 41 | num_Tos | Number of TOS |
| 42 | tcpcaplen_in_mean | Average size of the TCP header flowing into the packet |
| 43 | tcpcaplen_out_mean | Outgoing packet TCP header average size |
| 44 | tcpcaplen_in_max | Flow into the packet TCP header size maximum |
| 45 | Tctcpcaplen_out_max | Outgoing packet TCP header size is the largest |
| 46 | tcpcaplen_in_var | Flow into the packet TCP header square size |
| 47 | tcpcaplen_out_var | Outbound packet TCP header square size |
| 48 | size_seq_mean | Flow into the packet TCP serial number average |
| 49 | size_seq_max | Flow into the packet TCP serial number maximum |
| 50 | size_seq_min | Flow into the packet TCP sequence number minimum |
| 51 | size_seq_var | Flow into the packet TCP serial number square mean |
| 52 | ecn_in_pps/(ecn_in_pps + ack_in_pps) | |
| 53 | ecn_in_pps/(urg_in_pps + ack_in_pps) | |
| 54 | In_pps | Number of packets flowing per second |
| 55 | Out_pps | Number of packets flowing out per second |
| 56 | In_pps/(Out_pps + in_pps) | |
| 57 | size_in_pps | The total size of the packets flowing in per second |
| 58 | size_out_pps | The total size of packets flowing out per second |
| 59 | size_in_max | Maximum packet size per second |
| 60 | size_out_max | Maximum packet size per second |
| 61 | size_in_min | Minimum packet size per second |

**Table A1.** *Cont.*

| No. | Feature | Description |
| --- | --- | --- |
| 62 | size_out_min | Minimum packet size per second |
| 63 | size_in_mean | Average packet size per second |
| 64 | size_out_mean | Average packet size per second |
| 65 | size_in_var | Packet size average per second |
| 66 | size_out_var | Packet size average value per second |
| 67 | size_in_median | Median size of packets flowing in per second |
| 68 | size_out_medin | Median packet size per second |
| 69 | size_in_14 | Flows into the packet size 1/4 bit per second |
| 70 | size_out_14 | Outgoing packet size 1/4 bit per second |
| 71 | size_in_34 | Streaming packet size 3/4 min per second |
| 72 | size_out_34 | Outgoing packet size 3/4 min per second |
| 73 | port_in_size | Flow into port size |
| 74 | port_out_size | Outgoing port size |
| 75 | ttl | Whether the IP header TTL value changes |
| 76 | Duration | Duration of a flow |
| 77 | Interval_in_max | Streaming packet interval maximum |
| 78 | Interval_in_min | Streaming packet time interval minimum |
| 79 | Interval_in_mean | Average time interval of incoming packets |
| 80 | size_in_mean | Average packet size |
| 81 | size_out_mean | Average size of outgoing packets |
| 82 | Interval_in_var | Flow-in packet time interval mean |
| 83 | Interval_out_max | Outgoing packet interval maximum |
| 84 | Interval_out_min | Outgoing packet interval minimum |
| 85 | Interval_out_mean | Outgoing packet interval average |
| 86 | Interval_out_var | Mean time interval of outgoing packets |
| 87 | Interval_out_14 | Outgoing data packet interval 1/4 digit |
| 88 | Interval_out_34 | Outgoing packet interval 3/4 quart |
| 89 | payloadsize_in_max | Flow in valid data maximum |
| 90 | payload_out_max | Outgoing valid data maximum |
| 91 | payload_in_min | Flow into valid data minimum |
| 92 | payload_out_min | Outflow valid data minimum |
| 93 | payload_in_mean | Flowing in the effective data average |
| 94 | payload_ou_mean | Average value of outflow valid data |
| 95 | payloadsize_in_var | Average value of flowing data |
| 96 | payload_out_var | Mean value of outflow valid data |
| 97 | payload_in_14 | Flow in valid data 1/4 is divided into values |
| 98 | payload_out_14 | Outflow valid data 1/4 is divided into values |
| 99 | payload_in_34 | Flow in valid data 3/4 is divided into values |
| 100 | payload_out_34 | Outflow valid data 3/4 is divided into values |
| 101 | payload_in_median | Flow in valid data median value |
| 102 | payload_out_media | Outgoing valid data median value |

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
