# Peer review of "A Feature Analysis Based Identifying Scheme Using GBDT for DDoS with Multiple Attack Vectors"

_applsci, doi:10.3390/app9214633_

Round 1

Reviewer 1 Report

This paper investigates the detection of composite-type DDoS attacks. The authors first provide a comparison of different machine learning techniques from the perspective of their accuracies and running times. Based on this comparison, "gradient boosting decision tree" technique is selected for further optimizations. In particular the authors propose some parameter tuning techniques for feature selection. The optimizations improve the performance. A number of tables provide a detailed view of the performance improvements and the authors also provide an obtained attack vector feature tree.

The approach presented in the paper is interesting and I think it would be practically useful. The following is a list of issues that I hope the authors can take into account in a revision.

1) I think it may be useful to explain "composite-type" attacks in detail. Are there any useful references that explain such attacks and their effects? It may be interesting to provide such references in the introduction.

2) The second paragraph of the Introduction mentions that "most researchers use existing features and datasets". What are the disadvantages of the methods proposed by those researchers? It is not immediately clear what is meant by "existing features".

3) In the second paragraph of Introduction, it is hard to understand the expression "and the algorithm and algorithm tuning simply approximate the obtained result."

4) At the end of the first paragraph of Section 2, the authors mention "C&C attacks". However, no additional explanation is given to such attacks. Potential readers may benefit from additional information.

5) Please explain what "known attacks" and "unknown attacks" mean in the 5th paragraph of Section 2.

6) The explanation of the use of Kafka tool for DDoS feature extraction is quite hard to follow. It may be useful to provide a short summary on page 7 before providing technical details.

7) The paper makes comparisons to KNN, SVM, BN, and MLP algorithms. Could you please provide some brief information about these algorithms? It is not clear if these algorithms can be optimized to perform better. Are there any hyper parameters than can be tuned/optimized? To make the comparisons more systematic, the selection of hyperparameters need to be discussed.

8) On page 8, the authors state "The reason for the decrease in accuracy may be due to the inaccurate direction of the flow during the feature extraction phase". Is there a systematic approach to check this hypothesis?

9) The statement "The smaller the test time is, the shorter the DDoS attack that can be detected, and the DDoS will be more threatening to the server" on page 9 is not clear.

10) The authors mention the works [47] and [58] on the use of random forests in DDoS feature selection. Could you explain what is different in this paper in comparison to those works?

11) In the part where Pearson correlation coefficient is discussed the authors use the letter "a" with standard font the describe a scalar. Please use mathematical notation with a nonstandard font for this symbol.

12) I am confused with Table 4. There seems to be an error. It seems that the top section "Predictive output" should be "Actual output".

13) Table 7 provides feature subsets selected with different methods. Could you comment on some of the overlapping and nonoverlapping features? Are the any intiutive explanations to some of these features?

14) It is not very clear why the authors use the expression "cognitive approach" in the title, as the word "cognitive" does not appear anywhere else in the paper.

15) The following is a list of minor issues:

- In the abstract, "algorithm is combined" can be changed to "algorithm is used".
- In the second paragraph of the Introduction, please rephrase the sentence "Based on the characteristics of 102 DDoS attacks based on the TCP protocol and 49 DDoS attacks based on the UDP protocol," as the phrase "based on" is used too many times.
- In Definition 1, please put comma after "a3" and "3" as well as after "..." notation.
- After equation (1), please remove space in "r epresents".
- Page 8, line 335, "datasets" should be "dataset's".
- The caption of Figure 4 should be "Running time comparison".
- In the title of Section 6, "And" should be "and".
- In the definition of accuracy rate in (6), perhaps "PR" should be "AR".
- The title of Subsection 6.4 has an unnecessary "I".
- In Section 6.5.1, the font for the letter rho_0 does not math with rho in the caption of Figure 9.
- Page 14, line 531: put space in "at0.95".
- Page 16, line 541: remove the unnecessary period in "Figure. 10".
- Please sort the feature indices given in Table 8.
- Page 17, line 588: "each datasets" should be "each dataset".
- Page 17, line 593: put space in "tothe".
- Spacing and author name abbreviations in the references have problems.

Author Response

Thank you for taking the time to review my paper and give valuable advice. We have made corresponding changes and answers based on your comments, hoping to get your approval.

Reviewer 2 Report

• The abstract should be shorter... for instance, the specific contributions of the paper could be placed in a separated section, and only a short resume of those should be on the abstract.
• The paper presents a very interesting and relevant topic in the area of information security, since DDoS attacks are still one of the biggest challenges for security and availability of systems today.
• The paper is well written and well organised and the references are appropriated in number and relevance.
• The comparative analysis conducted by the authors about the different machine learning approaches to solve the DDoS problem is also very complete and interesting.
• The authors should highlight the name of tools, like PCAP, to differentiate from the rest of the text.
• I believe the authors are only considering types of DDoS attacks that are provoked through loads of internet traffic - it could be interesting also to consider other classes of attacks that are not based on the high loads of TCP or UDP requests, but that affect system vulnerabilities for crashing processes, for instance.

Author Response

(The authors gave the same response as above.)

Reviewer 3 Report

The paper is poorly written, which makes it hard to read.

How accurate is the data’s labels?

“This paper uses GBDT, KNN, SVM, BN, and MLP as the five machine learning algorithms to classify 1k TCP and 1k UDP 329 datasets. The results show that when classifying TCP traffic, the training accuracy of the GBDT algorithm reaches 0.99, and the test accuracy rate reaches 0.98.” => How was the data divided into train and test data? What does the accuracy measurement means here? Did the authors use the equation provided in line 492?

“As the size of the datasets increases, the accuracy of all algorithms decreases. …The reason for the decrease in accuracy may be due to the inaccurate direction of the flow during the feature extraction phase.” More explanation would be helpful?

Is the dataset used in Section 4 is same as the one used in Section 6?

Why the specification of the computer used in the experiment is described in Section 6 but not in Section 4?

What is the rational of using 210710281400.pcap data package only?

Line 498: Precision rate not accuracy rate?

Will ρ0 value of 0.6 for TCP and 0.3 UDP generalize well for other data files in MAWI?

Figure 10: The improvement achieved by using feature selection is not significant.

Author Response

(The authors gave the same response as above.)

Reviewer 4 Report

This paper is quite well written and organized and the reported experimental results are interesting.

However  I have some comments to do:

Authors should better show how this work advances the state of the art. The Authors should better investigate the SoA. There are several typos throughout the text that must be corrected The references must be correctly formatted (e.g. 53. 57.)

Author Response

(The authors gave the same response as above.)

Round 2

Reviewer 1 Report

The authors have addressed all of my comments and I think the overall presentation has improved. My assessment is positive. I hope the following minor point can be taken into account in the final version:

- Page 6, line 258, remove unnecessary space after "subscribed".
- Page 6, line 259, "kafka" should be "Kafka".

Reviewer 3 Report

It may update the paper to include your response.